# Rapid Identification of Pollen- and Anther-Specific Genes in Response to High-Temperature Stress Based on Transcriptome Profiling Analysis in Cotton

**DOI:** 10.3390/ijms23063378

**Published:** 2022-03-21

**Authors:** Rui Zhang, Lili Zhou, Yanlong Li, Huanhuan Ma, Yawei Li, Yizan Ma, Rongjie Lv, Jing Yang, Weiran Wang, Aierxi Alifu, Xianlong Zhang, Jie Kong, Ling Min

**Affiliations:** 1Key Laboratory of Crop Genetic Improvement, Huazhong Agricultural University, Wuhan 430070, China; ruizhang@webmail.hzau.edu.cn (R.Z.); lilizhou0730@outlook.com (L.Z.); yanlongli8@163.com (Y.L.); hhm@webmail.hzau.edu.cn (H.M.); liyaweia@126.com (Y.L.); mayizan@webmail.hzau.edu.cn (Y.M.); hunanlrj2021@163.com (R.L.); xlzhang@mail.hzau.edu.cn (X.Z.); 2Institute of Economic Crops, Xinjiang Academy of Agricultural Sciences, Urumqi 830091, China; hzjingy@126.com (J.Y.); xaaswwr@163.com (W.W.); xaasalifu@163.com (A.A.)

**Keywords:** *Gossypium hirsutum*, high-temperature stress, male-sterility, pollen-specific genes

## Abstract

Anther indehiscence and pollen sterility caused by high temperature (HT) stress have become a major problem that decreases the yield of cotton. Pollen- and anther-specific genes play a critical role in the process of male reproduction and the response to HT stress. In order to identify pollen-specific genes that respond to HT stress, a comparative transcriptome profiling analysis was performed in the pollen and anthers of *Gossypium hirsutum* HT-sensitive Line H05 against other tissue types under normal temperature (NT) conditions, and the analysis of a differentially expressed gene was conducted in the pollen of H05 under NT and HT conditions. In total, we identified 1111 pollen-specific genes (PSGs), 1066 anther-specific genes (ASGs), and 833 pollen differentially expressed genes (DEGs). Moreover, we found that the late stage of anther included more anther- and pollen-specific genes (APSGs). Stress-related *cis*-regulatory elements (CREs) and hormone-responsive CREs are enriched in the promoters of APSGs, suggesting that APSGs may respond to HT stress. However, 833 pollen DEGs had only 10 common genes with 1111 PSGs, indicating that PSGs are mainly involved in the processes of pollen development and do not respond to HT stress. Promoters of these 10 common genes are enriched for stress-related CREs and MeJA-responsive CREs, suggesting that these 10 common genes are involved in the process of pollen development while responding to HT stress. This study provides a pathway for rapidly identifying cotton pollen-specific genes that respond to HT stress.

## 1. Introduction

Temperature increases related to global warming reduce global yields of major crops [1,2,3,4]. It was estimated that global yields of wheat declined by 6.0%, rice by 3.2%, maize by 7.4%, and soybean by 3.1% for each degree Celsius increase in global mean temperature [5]. Cotton is cultivated in summer and is commonly affected by high temperatures (HT). Male sterility caused by HT in cotton has been reported in recent years, which has led to yield reductions [6,7,8,9]. Critical genes in response to HT in the male reproductive organ of cotton are poorly understood. Thus, finding pollen-specific genes and anther-specific genes involved in the HT response is necessary for HT tolerance cotton breeding.

In flowering plants, male reproductive development occurs in anthers and undergoes a series of complicated processes [10,11]. HT usually causes anther indehiscence and pollen abortion in cotton by impairing the process of microspore development. The tetrad stage was reported to be susceptible to heat stress, which generates germination-defective pollen, thus leading to male sterility [12]. Tapetum provides cellular contents and nutrients supporting pollen development, and it was reported to be sensitive to environmental stress: environmental stress causes early degeneration of tapetum tissues, leading to microspore abortion [13,14,15]. The flowering stage was reported to be susceptible to heat stress, which induces poor anther dehiscence [16,17,18]. In addition, three critical stages in the cotton anther development process were reported to be vulnerable to HT, including the tetrad stage, tapetum degradation stage and anther dehiscence stage [7,19]. The bud length of cotton was used for identification of the anther development stage, and buds with lengths of 6–7 mm, 9–14 mm, and >24 mm (or 19–24 mm) were divided into the tetrad stage, tapetum degradation stage, and anther dehiscence stage [7,20,21]. In recent years, the mechanisms underlying heat stress and male sterility have been revealed. The disordered accumulation of energy substances, such as lipids and sugars, and disrupted epigenetic modifications generally result in male sterility under HT conditions [12,19,22,23,24,25]. Downregulation of the auxin synthesis genes in the male reproductive organ caused by HT leads to pollen abortion, and male sterility can be rescued by exogenous auxin application [26,27]. Similarly, JA reduction induced by HT resulted in male sterility, and exogenous methyl jasmonate application in the early stages of anthers could reduce HT-driven male sterility [28]. In addition, a number of male-sterility genes induced by HT were found, such as *TT1* in rice, *ERECTA* in Arabidopsis, and *GhCKI* and *GhHRK1* in cotton [2,7,9,29].

Promoters, a *cis*-regulatory sequence, control development, and physiology by regulating gene expression [30]. Accurate and specific expression of anther-specific genes in the different stages of anther development is regulated by anther-specific promoters [31]. Pollen development from meiosis until mature pollen is a series of rapid changes in chromatin organization through different histone modifications [32,33]. Using anther-specific promoters to control and regulate the expression of anther-specific genes is a useful method to create male sterility lines or change fertility in crops. Recently, many anther-specific genes were identified in various plants by using microarrays and RNA-seq. In rice, 261 genes were identified as late-stage anther-specific genes that showed a 500-fold change in expression compared with other organs; six late-stage anther-specific genes were confirmed to be specifically expressed in anthers by using GUS staining [34]. Moreover, 627 later pollen-specific genes were identified in rice by comparing the transcriptomes of sporophytes and male gametes over time, eight of which were strongly expressed in the tricellular mature pollen stage [35]. By using RNA-seq, 269 late-stage anther-specific genes were identified in maize, and 13 promoters were confirmed to drive the α-amylase gene to disrupt pollen [36]. By comprehensive screening of the gene expression profile database in rice, 38 anther-specific genes were identified, of which 7 of the 38 promoters were confirmed to be specifically expressed in the later stage of anther development and could induce clear dominant male sterility [37].

The male reproductive organs of cotton are vulnerable to HT, which usually induces anther indehiscence and pollen abortion leading to male sterility. However, little is known about the anther- and pollen-specific genes in response to HT stress. In order to identify anther- and pollen-specific genes that respond to HT stress, we performed a comparative transcriptome profiling analysis in the pollen and anthers of *Gossypium hirsutum* HT-sensitive line H05 against other tissue types under normal temperature (NT) conditions, and we conducted a differentially expressed gene analysis in the pollen of H05 under NT and HT conditions. The screened candidate genes were analyzed and compared in predicted function and *cis*-acting elements to understand their differences. This research provides useful methods for the rapid identification of pollen-specific genes that respond to HT and identifies critical genes for HT tolerance breeding.

## 2. Results

### 2.1. Cotton Pollen Possesses a Unique Transcriptome

RNA-seq is a useful method to detect expression levels that are more precise and sensitive. In order to identify pollen-specific genes (PSGs) and anther-specific genes (ASGs), we conducted RNA-seq of 22 tissues of cotton thermosensitive Line H05 under normal temperature (NT) conditions, including mature pollen grains from anthers at anthesis, 13 different developmental stages anthers from the 3–5 mm, 5–6 mm, 6–7 mm, 7–8 mm, 8–9 mm, 9–10 mm, 10–11 mm, 11–12 mm, 12–13 mm, 13–14 mm, 14–16 mm, 16–19 mm, and 19–24 mm buds (bud length: from the nectary to the top of the bud) (Figure 1A), root, stem, leaf, bract, sepal, petal, stigma, and ovule. The relationship between the bud lengths and the anther/pollen developmental stages was reported in our previous study [21]. The microspore mother cells presented in the anther with bud lengths 3–5 mm, then the meiosis appeared in the microspore mother cells in anther with bud lengths 5–6 mm. Then the meiosis was completed, and the tetrads appeared in anther with bud lengths 6–7 mm. In the anther with bud lengths 7–8 mm, the microspores were released from the tetrad and then developed into uninucleate microspores (8–9 mm buds) and binucleate microspores (9–10 mm buds), then the mature pollen was generated in the anther with bud lengths 19–24 mm or more than 24 mm (anther dehiscence stage). After sequencing, raw data were cleaned and then mapped to the TM-1 reference genome [38]. Finally, we obtained the expression profile of 22 sequenced tissues (Appendix A).

Then, we performed a Pearson correlation coefficient analysis of transcript abundance to test the correlation of 22 sequenced tissues. The results showed that the transcripts of pollen have a strong correlation with anthers from buds with lengths higher than 8 mm and have a low correlation with other tissue types (Figure 1B). In particular, the transcripts of pollen clustered with anthers from 16–19 mm and 19–24 mm buds as a result of later stage anthers already having mature pollen grains. Therefore, we selected pollen and other tissue types except for anthers for PSG analysis; for the same reason, we used anthers and other tissue types except for pollen for ASG analysis.

In order to evaluate the transcript abundance in each of the sequenced tissues, we chose genes whose TPM (transcripts per kilobase of exon model per million mapped reads) value was higher than 1.00 as expressed genes (EGs), and we divided the EG into four groups: low expressed genes (LEGs, 1–10 TPM), medium expressed genes (MEGs, 10–30 TPM), high expressed genes (HEGs, 30–300 TPM), and very high expressed genes (VHEGs, ≥300 TPM) [39,40]. Of the 70,199 genes reported or predicted in the Gossypium cotton TM-1 reference genome, we found that pollen grains only have 12,005 EGs occupying 17.1% of known cotton genes (Table 1). Other tissues expressed 44.0–61.1% cotton genes with 30,917 to 42,867 EGs (Table 1). On the other hand, the average transcription of EGs in pollen was 82.8 TPM, while the average transcription of other tissues was less than 32.2 TPM (Table 1). Moreover, the percentages of top 300 EGs and top 1000 EGs in the transcriptome of pollen were 75.1% and 89.9%, respectively, whereas the percentages in the transcriptome of other tissues were 19.1–45.4% and 32.4–61.6%, respectively (Table 1). These results showed that cotton pollen had reduced expression of genes compared to other tissues, and the expression levels of genes were higher than those in other tissues, indicating that the maturation of pollen requires higher expression and specialized genes to be finished.

Furthermore, to understand the distribution of transcriptional levels, we counted the gene numbers of four transcript levels classed by abundance. We found that pollen had 449 VHEGs, slightly more than the other eight tissues (except anthers), which had 379 VHEGs on average (Figure 2A). However, pollen possessed only 8472 LEGs, 1633 MEGs, and 1415 HEGs, which significantly decreased compared to the average of the other eight tissues (except anthers), which had 22,471 LEGs, 9210 MEGs, and 5711 HEGs. These results indicated that cotton pollen had a lower transcriptome complexity than other tissues.

Pollen-specific genes were termed preferentially expressed in pollen or not expressed in other tissues except pollen. Thus, the criterion was used for screened PSGs as reported: (1) a TPM of at least 1 in mature pollen, and (2) expression at least 100-fold higher in pollen than in other vegetative or reproductive tissues [40]. In total, 1111 PSGs were identified (Appendix A). Moreover, 1111 PSGs occupied 9.3% of 12,005 cotton pollen EGs, and of these, all of them were predicted to encode proteins; 936 genes had functional descriptions, whereas the other 175 genes encoded uncharacterized proteins (Appendix A).

In order to understand the dependent relationship of gene expression between pollen and the other eight tissues, a two-dimensional scatter plot analysis was conducted. We used pollen expression values against each of the other eight tissues (except anthers) and found that pollen displayed a low correlation with each of the other eight tissues for gene expression, with R^2^ values less than 0.001 (Appendix A). Thus, we displayed the expression of PSGs in pollen against the maximum of the other eight tissues (Figure 2B). The PSGs are shown by blue points focused on the top left of Figure 2B, implying that PSGs were preferentially expressed in pollen instead of in other tissues. In addition, 449 pollen-abundant genes (PAGs) were identified by the criterion that the TPM value exceeded 300 and are shown by blue or red points above the red line (Figure 2B). In the 449 PAGs, 323 genes occupied 71.9% of which also belonged to PSGs (Figure 2B). In conclusion, these results demonstrated that cotton pollen has a special and unique transcriptome.

### 2.2. Cotton Anther Express a Transformable Transcriptome

In order to identify anther-specific genes in different developmental stages of H05 anthers under NT conditions, we collected and sequenced 13 stages of anthers depending on the length of the bud, including 3–5 mm, 5–6 mm, 6–7 mm, 7–8 mm, 8–9 mm, 9–10 mm, 10–11 mm, 11–12 mm, 12–13 mm, 13–14 mm, 14–16 mm, 16–19 mm, and 19–24 mm (Figure 1A). In this study, we used the same criterion to divide three critical developmental stages of anthers as we described before [7,19,21], including the tetrad stage (TS, anther in 6–7 mm bud including the tetrads), tapetal degradation stage (TDS, anther in 9–14 mm bud including uninucleate and binucleate microspores), and anther dehiscence stage (ADS, anther in 19–24 mm and ≥24 mm bud including the mature pollen grains) (Figure 1A).

The Pearson correlation coefficient analysis found that anthers from 8–24 mm buds have a low correlation with the other eight tissue types (except pollen) but have a strong correlation between themselves, indicating that adjacent stages of anthers have a similarity (Figure 1B). On the other hand, anthers from 16–19 mm buds clustered with anthers from 19–24 mm buds, and anthers from 8–16 mm buds clustered together. However, anthers in 3–5 mm, 5–6 mm, and 6–7 mm buds displayed indistinctive correlations with the other anthers and the other eight tissue types (Figure 1B). These results demonstrated that the early stage of anthers has a unique transcriptome, but TDS and the late stage of anthers have a similarity.

EGs were filtered by TPM values over 1.00 and classified into four groups: LEGs, MEGs, HEGs, and VHEGs. We found that the number of EGs in anthers from 6–7 mm buds was 42,867, which was higher than that in any other stages of anther, but the average transcription of EGs in anthers from 6–7 mm buds was less than that in any other anther stage (Figure 2C). Interestingly, the number of EGs decreased in anthers as the length of the bud increased from 6–7 mm to 16–19 mm. However, the number of EGs and the percentage of the top 300 and top 1000 EGs increased in anthers as the length of bud increased from 6–7 mm to 16–19 mm. Moreover, the number of LEGs, MEGs, and HEGs also decreased, but the number of VHEGs increased in anthers as the length of the bud increased (Figure 2C). These results indicated that the development of anthers and the maturation of pollen require more VHEGs and suppress the number of genes that are relatively expressed at low levels.

To identify anther-specific genes (ASGs) in different developmental stages, we used the average TPM value of 13 sequenced anthers for analysis. The criteria used to identify ASGs were the same as PSGs. In total, we identified 1066 ASGs (Appendix A). Moreover, we also identified 862 TS-specific genes (TSSGs), 950 TDS-specific genes (TDSSGs), and 1112 ADS-specific genes (ADSSGs) using the data of anthers in 6–7 mm buds including the tetrads, anthers in 11–12 mm buds including binucleate microspores, and anthers in 19–24 mm buds including the mature pollen grains (Appendix A). All 1066 ASGs were predicted to encode proteins: 880 genes had functional descriptions, but the other 186 genes encoded uncharacterized proteins. To confirm that ASGs were preferentially expressed in the different developmental stages of anthers, we randomly selected 16 ASGs and performed RT-PCR and qRT-PCR analysis in 21 tissues of H05. The results showed that 5 of 16 selected ASGs were preferentially expressed in TS, 3 of 16 selected ASGs were preferentially expressed in TDS, 3 of 16 selected ASGs were preferentially expressed in ADS, and the remaining 5 were preferentially expressed in anthers in 5–6 mm or 7–8 mm buds (Figure 3B and Appendix A). In addition, the RT-PCR and RNA-seq results exhibited high similarity, and the Pearson correlation analysis indicated excellent concordance between the RNA-Seq and qRT-PCR results (Figure 3A,B and Appendix A). Thus, these results confirmed that 1066 ASGs were real specific genes in anthers at different developmental stages.

Moreover, we obtained a genome editing line of *Ghir_D12G012350,* which is one of the 16 candidates ASGs identified by RT-PCR and qRT-PCR (Figure 4A). The mutant of *Ghir_D12G012350* exhibited no significant defects in vegetative growth, but the anthers of mutant showed poor dehiscence compared to WT (Jin668) under NT conditions (Figure 4B,C). This result showed that the loss function of *Ghir_D12G012350* impacted cotton anther development and confirmed our method to identify ASGs were practicable.

In addition, we performed a two-dimensional scatter plot analysis of transcripts using the average TPM value of anthers against the other eight tissues (except pollen) (Figure 2D). The results showed that ASGs were gathered on the top left in Figure 2D, which indicated that ASGs were specifically expressed in anthers but not in other tissues. We identified 362 anther-abundant genes (AAGs) with transcription levels over 300 TPM (Appendix A). In 362 AAGs, 99 or 27.3% of them were also ASGs (Figure 2D).

### 2.3. The Function of Anther- and Pollen-Specific Genes (APSGs) Are Predicted to Be More Like PSGs

The overlapping genes in ASGs and PSGs are called anther- and pollen-specific genes (APSGS). Using the online software Venny 2.1.0 (Venny 2.1.0), we identified 294 APSGs (Figure 5A, Appendix A). As a result of anthers expressing a transformable transcriptome with the development of anthers and maturation of pollen, we also counted the number of TS- and pollen-specific genes (TSPSGs), TDS- and pollen-specific genes (TDSPSGs), and ADS- and pollen-specific genes (ADSPSGs). In total, 39 TSPSGs, 327 TDSPSGs, and 742 ADSPSGs were identified (Figure 5B–D). The results showed that the number of APSGs remarkably increased with the development of anthers and maturation of pollen, suggesting that a later stage of anthers included more pollen-specific genes.

In order to understand the potential function of PSGs, ASGs and APSGs in anther and microspore development, anther dehiscence, and pollen maturation, we obtained a functional description of PSGs, ASGs, and APSGs from CottonFGD (CottonFGD: HomePage) (Appendix A). We further performed gene ontology (GO) enrichment analysis of 817 PSGs, 772 ASGs, and 294 APSGs. A total of 817 PSGs and 772 ASGs were the remaining genes of 1111 PSGs and 1066 ASGs, except 294 APSGs. The results showed that 294 APSGs were assigned to 14 GO terms (Figure 6A). We also classified 14 GO terms into three groups: biological process, cellular component, and molecular function. For GO terms of the biological process, APSGs were only in three biological processes: the cell wall modification, clathrin coat assembly, and actin filament bundle assembly. For GO terms of the cellular component, APSGs were only in two molecular functions: the cell wall and clathrin-coated vesicle. For GO terms of the molecular function, APSGs were in nine cellular components: the enzyme inhibitor activity, pectinesterase activity, copper ion binding, pectate lyase activity, polygalacturonase activity, carbohydrate binding, clathrin binding, phospholipid binding, and 1-phosphatidylinositol binding (Figure 6A). On the other hand, 39 GO terms were identified in the 817 PSGs, whereas only 6 GO terms were identified in the 772 ASGs (Figure 6B,C). In the 39 GO terms of 817 PSGs, 13 GO terms were assigned to the biological process, mainly including the cation transport, signal transduction, cell wall modification, and clathrin coat assembly; 6 GO terms were assigned to the cellular component, mainly including the cell wall and clathrin-coated vesicle; and 20 GO terms were assigned to the molecular function, mainly including the enzyme inhibitor activity, solute and pectinesterase activity (Figure 6B). In the six GO terms of 772 ASGs, only lipid transport was assigned to the biological process; no GO term was assigned to the cellular component; and five GO terms were assigned to the molecular function, mainly carbohydrate binding, polygalacturonase activity, and lipid binding (Figure 6C). Interestingly, we also found 10 of 14 GO terms in the APSGs in PSGs. However, only 2 of 14 GO terms in the APSGs belonged to ASGs, and there were no same GO terms both in PSGs and ASGs. These results suggested that the function of APSGs was more similar to that of PSGs, not ASGs.

### 2.4. Stress-Related and Hormone-Responsive Cis-Elements Enrich in the Promoters of APSGs

To reveal the function of *cis*-regulatory elements (CREs) of APSGs in the expression and regulation of APSGs, we performed a CRE analysis of the 2-kb promoter region upstream of the start codon (ATG) of 294 APSGs via PlantCARE [41]. We identified 116 known CREs and 13 unnamed CREs of 294 APSGS promoters in total (Appendix A). The top 10 most frequent of 116 CREs are listed as follows: TATA-box, CAAT-box, Box4, MYC, MYB, ERE, ARE, GT1-motif, ABRE, and STRE. Interestingly, we identified a temperature responsiveness *cis*-acting element (TR) in 131 APSGS promoters, which suggested that TR may regulate the expression of these 131 APSGs in response to HT stress.

In order to understand the function of 116 CREs, we manually classified 116 CREs into 11 groups based on functional annotation. The 11 groups contained common CERs (TATA-box and CAAT-box), tissue-specific CREs, light-responsive CREs, stress-related CREs, temperature-responsive CREs, MeJA-responsive CREs, ABA-responsive CREs, SA-responsive CREs, GA-responsive CREs, auxin-responsive CREs, and other CREs.

Furthermore, we calculated the frequency (average sites per promoter) of the 11 CREs (Figure 7). The results showed that the common CREs TATA-box and CAAT-box were most enriched in the promoter of 294 APSGs. The frequencies of TATA-box and CAAT-box were 63.68 and 40.83 per promoter of APSGs, respectively (Figure 7). In addition to common CREs, stress-related CREs were also enriched, and temperature-responsive CREs could be found in APSGS promoters in HT-sensitive Line H05, which indicated that APSGs of H05 might be susceptible to the environment. In other words, each promoter of APSGs had at least 20 copies of stress-related CREs and approximately 1 copy of temperature-responsive CREs (Figure 7). Moreover, each promoter of APSGs had 3 copies of tissue-specific CREs and 12 copies of light-responsive CREs, suggesting that these two CREs are important factors for APSG expression (Figure 7). In addition, five hormone-responsive CREs were also found in the promoters of APSGs. Compared with the other four hormone-responsive CREs, MeJA-responsive CREs were more enriched in the promoter of APSGs, with 3.41 copies per APSGS promoter, which indicated that jasmonic acid (JA) metabolic pathways are involved in the regulation of APSGS expression (Figure 7). Together, these results showed that the expression and regulation of APSGs respond to HT stress require stress-related and hormone-responsive *cis*-acting elements.

### 2.5. PSGs Mostly Do Not Respond to HT and Have Distinct Differences from Pollen Differentially Expressed Genes (DEGs)

In order to identify pollen-specific genes in response to HT, we also sequenced pollen of HT-sensitive Line H05 under HT conditions. After a series of RNA-seq data analyses as we described above, we obtained the expression profiles, including the count value and TPM value, of pollen from anthers at anthesis under NT conditions (NTpollen) and HT conditions (HTpollen) (Appendix A). The count values of NTpollen and HTpollen were used to identify differentially expressed genes (DEGs) via edgeR software [42]. The BCV value (square-root-dispersion) was set as 0.3, and other parameters were set as default [42].

In total, we identified 833 DEGs, including 162 up-regulated genes and 671 down-regulated genes (Figure 8A,B, Appendix A). We found that the number of up-regulated genes was significantly lower than the number of down-regulated genes (Figure 8B). This result suggested that HT reduces the activity of pollen by suppressing the gene expression level. Moreover, we tested the relationship between DEGs and PSGs using the online software Venny 2.1.0. The results showed that only 10 DEGs belonged to APSGs, and no DEGs belonged to PAGs, indicating that the genes in response to HT stress were mostly not pollen-specific genes (Figure 8C). We further investigated the relationship between DEGs and four types of EGs. The results are listed as follows: 22 up-regulated genes and 427 down-regulated genes occupied 53.9% of DEGs belonging to pollen low expressed genes; only 1 up-regulated gene and 21 down-regulated genes of DEGs belonged to pollen medium expressed genes; 3 up-regulated genes and 9 down-regulated genes of DEGs belonged to pollen high expressed genes; and there was no DEGs belonged to pollen very high expressed genes (Appendix A). These results suggested that only a few pollen-specific genes respond to HT stress, and pollen differentially expressed genes under HT were distinct from pollen-specific genes.

In order to understand the function of pollen DEGs under HT, the functional description of DEGs from CottonFGD was downloaded, showing that 778 of 833 DEGs had a functional description, and 67 DEGs encoded uncharacterized proteins. We also found that approximately 40 DEGs encode heat shock proteins, suggesting that these DEGs respond to HT (Appendix A). To reveal the function of DEGs, we conducted a GO enrichment analysis. However, only 13 GO terms were enriched and divided into two groups: the biological process and molecular function (Figure 8D). For GO terms of the biological process, DEGs were only in four biological processes: the protein metabolic process, sucrose metabolic process, proton export across the plasma membrane, and anion transport (Figure 8D). For GO terms of the molecular function, 9 GO terms were identified: cation binding, channel activity, unfolded protein binding, sucrose synthase activity, oxidoreductase activity, proton-exporting ATPase activity, inorganic anion exchanger activity, fructose-bisphosphate aldolase activity, and alpha-amylase activity (Figure 8D). These 13 GO terms were enriched in DEGs totally different from APSGs and PSGs, which further suggested that pollen differentially expressed genes were distinct from pollen-specific genes and pollen-specific genes that did not respond to HT stress.

On the other hand, we obtained the functional description and expression value of the 10 genes that both existed in DEGs and PSGs (Appendix A). We found that only two genes had no functional descriptions and encoded uncharacterized proteins (Appendix A). The other eight genes that had functional descriptions were berberine bridge enzyme-like 17 (BBE17), probable pectinesterase inhibitor 21 (PME21), B3 domain-containing transcription factor VRN1 (VRN1), anther-specific protein LAT52 (LAT52), transcription factor HHO5 (HHO5), putative defensin-like protein 244 (SCRL11), nuclear transcription factor Y subunit B-9 (NFYB9), and polyphenol oxidase (Ghir_D06G019990). Moreover, these 10 genes were mostly preferentially expressed in pollen compared with other tissues (Appendix A). We further conducted a CRE analysis of the promoters of these 10 genes. The results showed that stress-related CREs and light-responsive CREs were enriched in the 10 gene promoters, indicating that these two CREs are important for resistance to HT stress (Figure 9). Temperature-responsive CREs were identified in the promoters of BBE17 and Ghir_D09G008810 (Figure 9). In addition, four types of hormone-responsive CREs were identified, and MeJA-responsive CREs were more enriched in the promoters of these 10 genes compared with ABA-, SA- and GA- responsive CREs, suggesting that JA plays a more important role in response to HT than other hormones (Figure 9).

Together, these results show that pollen DEGs have a distinct difference from PSGs, suggesting that pollen-specific genes are mostly involved in the process of pollen development rather than responding to HT stress.

## 3. Discussion

### 3.1. Transcriptome Diversity in Cotton Anthers and Pollen

In flowering plants, microspore development and pollen maturation occur in the loculi of anthers [11]. Based on cellular morphology, anther development of *Arabidopsis* was divided into 14 stages [43]. During stages 1 to 8, anther morphology was established, and microspore mother cells formed and then underwent meiosis. By the end of stage 8, specialized cells and tissues formed, including tetrads and microspores released from the tetrads. During stages 9 to 14, the young anthers grew consistently until pollen maturation. By the end of stage 14, mature pollen grains formed that were waiting to be released. Three early stages of microspore development in stages 1 to 8 and three stages of pollen maturation in stages 9 to 14 included archesporial cells, meiocytes, the quartet stage of microspore, uninucleate microspore, binucleate microspore, and mature pollen [10,11,43]. Epigenetic regulators play an important role in the process of pollen development by controlling pollen-specific gene expression through the chromosome. Histone modification and chromatin remodeling are involved in the regulation of pollen-specific genes that influence pollen development [32,33]. SDG701, a methyltransferase, was reported to be crucial for gametophytic transmission by specifically catalyzing H3K4 methylation [44]. SDG2-mediated H3K4me3 deposition was reported to be involved in epigenetic reprogramming during the process of male gametogenesis in Arabidopsis. Additionally, SDG2 participated in promoting chromatin decondensation in the pollen vegetative nucleus, which is critical for post-meiotic microspore development [45]. Cotton anther development undergoes similar stages as *Arabidopsis* and can be distinguished based on bud length [20,21]. In this study, to identify anther-specific genes in different developmental stages of anther, we selected 13 continuous stages of anthers from the 3–5 mm bud to 19–24 mm bud to sequence, which covers the most developmental stages of the cotton anther.

In maize, 44 k microarray transcriptome profiling identified more than 24,400 transcripts in each of the six anther developmental stages plus mature pollen. Maize anthers express varied transcripts with the development of anthers; in particular, the three early stages, A1.0 through A2.0 (entry into meiosis), each express more than 20,000 transcripts while the end of meiosis (Q stage) loss approximately 2700 transcripts compared with the three early stages [46]. The number of expressed genes identified in four stages of anther development in rice, including pre-meiotic, meiotic, anthers at single-celled, and tri-nucleate pollen, were 17,497, 18,090, 17,953, and 15,465, respectively, which revealed that meiotic pollen possesses the highest number of expressed genes, tri-nucleate pollen possesses the lowest number of expressed genes, and the number of expressed genes decreases with anther development [47]. Honys et al. (2004) found that the number of transcripts declined, and the proportion of male gametophyte-specific transcripts increased in the transition from bicellular to tricellular pollen in Arabidopsis [48]. In this study, we sequenced 13 anthers at different development stages, including TS, TDS, and ADS, according to the bud length. We found that the number of expressed genes varied in 13 anthers, TS (6–7 mm bud) expressed the maximum number of expressed genes, anthers from 16–19 mm buds possessed the smallest number of expressed genes, and the number of expressed genes consistently decreased in anthers between these two stages. However, the number of very high expressed genes was lowest in TS, highest in anthers from 16–19 mm buds, and showed an approximately continuous increase in these two stages.

Unique characteristics of the pollen transcriptome were identified in Arabidopsis: the number of expressed genes in pollen is approximately one-third of the number of other organs on average [49]. Transcriptome profiling based on RNA-seq revealed that mature pollen grains possess 9242 EGs, which is less than 30% of known maize genes, whereas other reproductive tissues and vegetative tissues express approximately 80% of maize genes, with 24,175 to 27,659 EGs [40]. The transcripts of the top 300 EGs and top 1000 EGs in pollen occupied 75.5% and 91.7% of the pollen transcriptome, while this percentage in other tissues was only were 28.2–47.2% and 43.4–69.2% [40]. In maize, microarray-based transcriptome profiling revealed that mature haploid pollen expresses approximately 10,000 transcripts, half of the less than 20,000 transcripts expressed in post-meiotic anthers [46]. In our study, we found that cotton pollen expresses 12,005 genes, obviously less than other tissues, with 30,917 to 42,867 EGs. However, the average transcription of EGs in cotton pollen was 82.2 TPM higher than for other tissues with 23.2 to 32.3 TPM, and the percentage of the top 300 EGs and top 1000 EGs in the transcriptome of pollen was largely higher than that in the other eight tissues. Moreover, pollen expresses 449 very high expressed genes (VHEGs), which is higher than the other eight tissues with 379 VHEGs, but pollen expresses largely reduced low expressed genes, medium expressed genes, and high expressed genes compared with other tissues. Our results are consistent with the studies in Arabidopsis, rice, and maize in which the cotton pollen transcriptome had reduced EGs but increased transcriptional levels, and the cotton anther transcriptome varied with the development of anthers while showing the characteristics of the pollen transcriptome.

### 3.2. The Late Stage of Cotton Anther Contains More Anther- and Pollen-Specific Genes

Genes specifically expressed in anthers or pollen have differences in the development of pollen and can be divided into early pollen-specific genes (from the archesporial cell-forming stage to uni-cellular gametophyte stage) and late pollen-specific genes (from the bi-cellular gametophyte stage to germinating pollen). Endo et al. (2014) identified 156 anther-specific genes in the late pollen stages by analyzing rice microarray data [50]. Akasaka et al. (2018) identified 38 anther-specific genes and further confirmed seven genes expressed specifically in anthers at later developmental stages [37]. Ma et al. (2008) identified 1952 genes specifically expressed in the early anther development stages (before meiosis) in maize [46]. Nguyen et al. (2016) identified 410 genes preferentially expressed in the early developmental stage of pollen in rice [51]. Moreover, 261, 627, and 269 late pollen-preferred genes were identified in rice [34,35,36].

Deveshwar et al. (2011) identified 1000 genes specifically expressed in rice anther stages and the proportions of anther-specific genes in the four stages of anther development, pre-meiotic, meiotic, anthers at single-celled and tri-nucleate pollen, were 0.3%, 0.5%, 2.0%, and 4.4%, respectively, which suggested that a late stage of anther includes more anther-specific genes and that the number of anther-specific genes increases with the development of anther [47]. Shi et al. (2021) identified 1215 mature pollen-specific genes using a more stringent 100-fold criterion, and 1009 mature anther-specific genes were identified in the late stage of anther development using the same criterion [40]. Moreover, similar GO terms were found in mature pollen-specific genes and mature anther-specific genes, and 623 shared genes (mature anther- and pollen-specific genes) in mature anther-specific genes and mature pollen-specific genes were also identified [40].

In this study, we identified anther-specific genes (ASGs) in different developmental stages of anthers, including the tetrad stage (TS, 6–7 mm bud), tapetal degradation stage (TDS, 9–14 mm bud), and near anther dehiscence stage (ADS, 19–24 mm bud). In total, we identified 862 TS-specific genes (TSSGs), 950 TDS-specific genes (TDSSGs), and 1112 ADS-specific genes (ADSSGs) with the same standard, and 1066 pollen-specific genes (PSGs). The numbers of these gene sets were indistinct differences. However, the numbers of TS- and pollen-specific genes (TSPSGs), TDS- and pollen-specific genes (TDSPSGs), and ADS- and pollen-specific genes (ADSPSGs) were 39, 327, and 742, respectively, which showed that the number of anther- and pollen-specific genes (APSGs) increased with the development of anthers and that the late stage of anthers included more APSGs. In addition, we used the average transcript value of 13 sequenced anthers against other tissues to identify ASGs, and 1066 ASGs were identified. Furthermore, using these 1066 ASGs and 1111 PSGs, we identified 293 APSGs. Interestingly, similar GO terms were found in APSGs and PSGs, indicating that 293 APSGs were mainly involved in the process of microspore development and pollen maturation. However, only two of the same GO terms were found in APSGs and ASGs, suggesting that the function of APSGs is different from that of ASGs. Our results indicated that the late stage of anthers contains more APSGs and that the function of APSGs is predicted to be more similar to PSGs than to that of ASGs.

### 3.3. Few Pollen-Specific Genes (PSGs) Are Involved in Response to High-Temperature (HT) Stress

Many pollen-specific genes were reported in various plant species, such as *RA8* in rice [52]; *NPT303* [53] in tobacco; *Zm58.1*, *ZmC5*, *MATRILINEAL* and *ZmSTK2_USP* in maize [54,55,56,57]; *LAT52* in tomato [58]; *TUA1* in *Arabidopsis thaliana* [59]; and *Bp4* in *Brassica napus* [60]. However, few genes in response to HT stress were reported.

In our previous study, early expression of *CASEIN KINASE I* (*GhCKI*) in an HT-sensitive cotton Line was induced by HT, causing male sterility [18]. The AtMYB24 transcription factor showed similar expression to *AtCKL2* and *AtCKL7*, which was induced early in the anther of Arabidopsis under HT stress [61]. Thermo-tolerance 1 was reported as a thermotolerance gene in rice that eliminates cytotoxic denatured proteins more efficiently and maintains heat-response processes more effectively to protect cells under heat stress [29]. Shen et al. found that *Arabidopsis thaliana* receptor-like kinase *ERECTA* (*ER*) is a positive regulator of the heat stress response and that overexpressed *ER* in Arabidopsis, rice, and tomato can enhance heat tolerance [2]. In our previous study, *GhHRK1* was confirmed to negatively respond to HT in both cotton and Arabidopsis: heterologous expression of *GhHRK1* in Arabidopsis decreased silique length, and *hrk1* mutants obtained more seeds than WT under HT conditions; misexpression of *GhHRK1* at the tetrad stage caused male sterility in HT-sensitive cotton accessions [9]. In this study, to understand the relationship between stress and male reproductive development, we conducted an analysis of differentially expressed genes in pollen under NT and HT conditions. In total, 833 differentially expressed genes (DEGs) were identified. However, we found that only 10 DEGs belonged to pollen-specific genes (PSGs), and 449 DEGs belonged to low expressed genes (LEGs) in cotton pollen. Among the 10 DEGs that respond to HT stress, LAT52 was reported as a tomato pollen-specific gene, which indicated that these 10 DEGs might be real pollen-specific genes. Our results suggested that most PSGs mainly involved the process of pollen development rather than in response to HT stress, and only a few PSGs involved these two processes. Thus, these 10 DEGs were confirmed to have functions in the cotton pollen response to HT.

## 4. Materials and Methods

### 4.1. Plant Materials and HT Treatment

The cotton HT-sensitive line H05 was sampled to perform RNA-seq, RT-PCR, and qRT-PCR experiments. The agronomical traits of cotton HT-sensitive line H05 were as follows: approximately 1.2 m height, loose plant type, medium leaves, infinite fruit branches, the fruit branches were in 8 to 11 plates, among which 6 to 8 plates were most likely to be sterile and shed under HT, and the growth period was about 130 days. As we described before, the anthers of H05 show normal dehiscence under NT conditions but exhibit indehiscence and partial male sterility under HT conditions, and the HT stress could cause pollen abortion of H05 [7,19]. The cotton Jin668 is the transgenic receptor material with an extremely high regeneration ability, and it was cultivated by successive regeneration acclimation of its maternal inbred Y668 cultivar. TM-1 is a standard reference cotton (*Gossypium hirsutum* L.) for genetic and cytogenetic experimentation.

The transgenic line and J668 (wild type) were cultivated in the greenhouse under NT conditions. The cotton (*Gossypium hirsutum*) HT-sensitive Line H05 was cultivated in the greenhouse. The conditions of 28–35 °C/20–27 °C day/night and 35–39 °C/29–31 °C day/night were set as normal temperature (NT) conditions and high temperature (HT) conditions, respectively. Anthers from lengths of 3–5 mm, 5–6 mm, 6–7 mm, 7–8 mm, 8–9 mm, 9–10 mm, 10–11 mm, 11–12 mm, 12–13 mm, 13–14 mm, 14–16 mm, 16–19 mm, and 19–24 mm buds (bud length: from the nectary to the top of the bud), root, stem, leaf, bract, sepal, petal, stigma, and ovule were sampled under NT conditions. Matured pollen grains at anthesis were collected under NT and HT conditions, respectively. All the samples were harvested and frozen in liquid nitrogen immediately or stored at −70 °C until use.

### 4.2. RNA-Seq, RT-PCR, and qRT-PCR

Total RNA was extracted using a modified guanidine thiocyanate method [18] and was sent to Novogene (Tianjin, China). At least 2 μg of RNA from each sample was used to construct standard Illumina transcriptome sequencing libraries. First, the mRNA was enriched using oligo (dT) magnetic beads, and the concentrated poly-A mRNA was digested into short fragments; then, the first- and second-strand complementary DNAs were synthesized using the mRNA fragments as templates. After purification and the addition of single nucleotide A (adenine), the short fragments were connected with adapters. The libraries were then sequenced using an Illumina HiSeqTM 4000 PE150 platform. Each sample obtained approximately 10 Gb clean bases. Adapters and low-quality reads were clipped using Trimmomatic software [62]. All remaining reads were mapped to the cotton reference genome using STAR software [63]. The transcript was quantified using RSEM [64]. Gene expression was calculated and subsequently normalized to TPM (transcripts per kilobase of exon model per million mapped reads) [39]. The results of RNA-seq were obtained from 1 biologically independent experiment.

In order to confirm the expression pattern of the anther-specific genes selected in the RNA-seq experiments, RT-PCR and qRT-PCR were performed. The same gene-specific primers were used for RT-PCR and qRT-PCR (Appendix A). qRT-PCR was performed using an ABI 7500 real-time PCR system. The cotton *GhUBIQUITIN7* gene was used as a control for qRT-PCR and RT-PCR. Relative gene expression levels for qRT-PCR were calculated using the 2^−ΔΔCt^ method as previously described [7]. The results of qRT-PCR were obtained from 3 biologically independent experiments. The samples used for qRT-PCR were different from those for RNA-seq. All the samples were obtained from the greenhouse plants.

### 4.3. Identification of PSGs, ASGs, APSGs, and Pollen DEGs in the Cotton Genome Using RNA-Seq Datasets

We performed RNA-seq analysis of 22 tissues under NT conditions and obtained the transcriptome profiles of these 22 tissues (Appendix A). The PSGs were identified according to transcript abundance ratio in pollen versus the maximum transcription abundance in the other eight tissues. The ASGs were identified by the ratio of the average transcript abundance in anthers versus the maximum transcription abundance in the other eight tissues. TS-specific genes (TSSGs), TDS-specific genes (TDSSGs), and ADS-specific genes (ADSSGs) were identified by the ratio of transcription abundance in anthers from 6–7 mm buds, 11–12 mm buds, and 19–24 mm buds versus the maximum transcription abundance in the other eight tissues. The threshold of the ratio was set at 100-fold to identify PSGs, ASGs, TSSGs, TDSSGs, and ADSSGs. Anther- and pollen-specific genes (APSGs) are the overlapping genes of ASGs and PSGs. TS- and pollen-specific genes (TSPSGs), TDS- and pollen-specific genes (TDSPSGs), and ADS- and pollen-specific genes (ADSPSGs) are the shared genes of PSGs with TSSGs, TDSSGs, and ADSSGs, respectively. EdgeR software was used to identify pollen differentially expressed genes (DEGs) with a cutoff *p*-value < 0.05 [42]. We trimmed the pollen data under NT and HT conditions according to the criterion of count value ≥ 1 in at least one sample. We tried different BCV values to identify DEGs, and we found that 0.3 was our best selection.

### 4.4. GO Enrichment and Cis-Elements Analysis of PSGs, ASGs, and APSGs

ClusterProfiler software was used to perform GO enrichment analysis, and significantly enriched GO terms were acquired with a cutoff *p*-value < 0.05 and a cutoff *q* value < 0.2 [65]. The 2-kb promoters were extracted from CottonFGD (CottonFGD: HomePage). The PlantCARE database was used to predict *cis*-regulatory elements (CREs) of each promoter with default parameters [41]. Predicted CREs were manually divided into 11 groups according to their function. The distribution of CREs was drawn by software TBtools [66].

## 5. Conclusions

In summary, we identified 1066 anther-specific genes and 1111 pollen-specific genes through comparing 22 differential tissue RNA-seq datasets of HT-sensitive Line H05, and we obtained 10 pollen-specific genes that respond to HT stress through comparing the datasets of 1111 pollen-specific genes and 833 pollen differentially expressed genes. Furthermore, through genome editing an anther-specific gene, we obtained mutant plants with indehiscent anthers to confirm the method. In addition, the GO enrichment analysis and *cis*-regulatory elements analysis of 10 pollen-specific genes suggested that they are involved in the process of pollen development while responding to HT stress. Together, this study provides a rapid pathway for identifying anther- and pollen-specific genes and provides genetic resources for breeding HT-tolerant cotton lines.

## Figures and Tables

**Figure 1 ijms-23-03378-f001:**
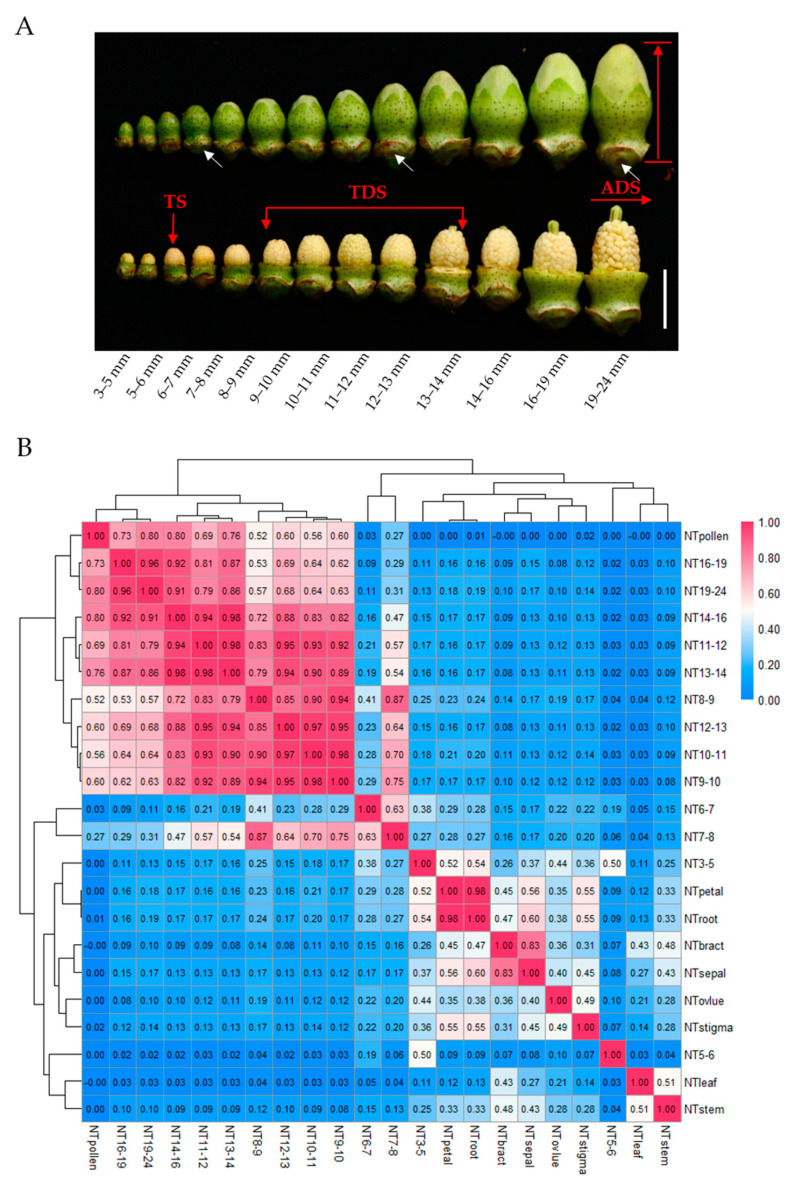
Phenotype of H05 anthers under Normal Temperature (NT) conditions and Pearson’s correlation coefficient r matrix of 22 sequenced tissues. (**A**) Phenotype of H05 anthers at different development stages under NT conditions. TS, tetrad stage (anther in 6–7 mm bud with tetrads). TDS, tapetal degradation stage (anther in 9–14 mm bud with uninucleate microspores or binucleate microspores). ADS, anther dehiscence stage (anther in 19–24 mm and ≥24 mm bud with mature pollen grains). The white arrow represents the nectary. Bud length: from the nectary to the top of the bud. Scale bars = 1 cm; (**B**) Pearson’s correlation coefficient r matrix of 22 sequenced tissues under NT conditions. TPM values for the transcripts of 70,199 genes were used to perform this correlation analysis. Pearson’s correlation coefficient r was calculated by R language and filled as numbers in this matrix. Pearson’s correlation matrix was drawn by heatmap software. Twenty-two sequenced tissues of H05: mature pollen at anthesis under NT (NTpollen), anther in 3–5 mm bud under NT (NT3-5), anther in 5–6 mm bud under NT (NT5-6), anther in 6–7 mm bud under NT (NT6-7), anther in 7–8 mm bud under NT (NT7-8), anther in 8–9 mm bud under NT (NT8-9), anther in 9–10 mm bud under NT (NT9-10), anther in 10–11 mm bud under NT (NT10-11), anther in 11–12 mm bud under NT (NT11-12), anther in 12–13 mm bud under NT (NT12-13), anther in 13–14 mm bud under NT (NT13-14), anther in 14–16 mm bud under NT (NT14-16), anther in 16–19 mm bud (NT16-19), anther in 19–24 mm bud (NT19-24), root under NT (NTroot), stem under NT (NTstem), leaf under NT (NTleaf), bract under NT (NTbract), sepal under NT (NTsepal), petal under NT (NTpetal), stigma under NT (NTstigma), and ovlue under NT (NTovlue).

**Figure 2 ijms-23-03378-f002:**
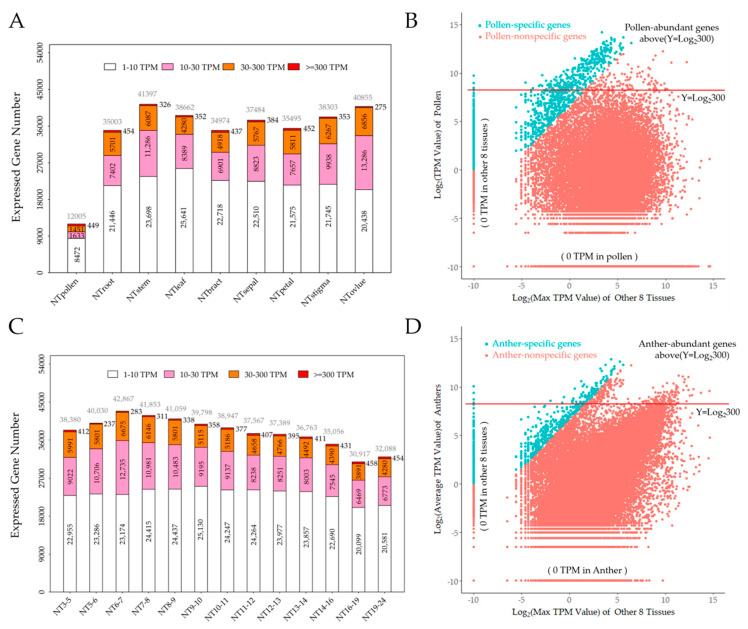
The distribution of transcriptional levels, PSGs and ASGs in H05 tissues. (**A**) The distribution of transcriptional levels grouped by abundances in the nine selected H05 tissues under NT conditions. Very highly expressed genes (VHEGs), highly expressed genes (HEGs), medium expressed genes (MEGs), and low expressed genes (LEGs) are indicated in dark red, orange, pink, and white color, respectively. VHEGs, HEGs, MEGs, and LEGs possess the transcript levels of ≥300, 30–300, 10–30, and 1–10 TPM, respectively. The total EG number of each tissue was indicated on the top of the columns in grey color. The transcript profiles of nine selected tissues were extracted from Appendix A; (**B**) Two-dimension scatter plot of transcript abundance in pollen vs. the max of other eight tissues under NT conditions. The x-axis indicates the max of transcript abundances in the other eight tissues (except pollen and anthers), and the y-axis indicates the transcript abundance detected in pollen, with each point representing a unique gene model. Blue points and orange points represent the pollen-specific genes (PSGs) and pollen-nonspecific genes, respectively. Pollen-abundant genes (PAGs) are located above the redline (y = log_2_ 300); (**C**) The distribution of transcriptional levels grouped by abundances in the 13 selected anthers of H05 under NT conditions. Very highly expressed genes (VHEGs), highly expressed genes (HEGs), medium expressed genes (MEGs), and low expressed genes (LEGs) are indicated in dark red, orange, pink, and white color, respectively. VHEGs, HEGs, MEGs, and LEGs possess the transcript levels of ≥300, 30–300, 10–30, and 1–10 TPM, respectively. The total EG number of each tissue was indicated on the top of the columns in grey color. The transcript profiles of 13 anthers were extracted from Appendix A; (**D**) Two-dimension scatter plot of transcript abundance in the average of anthers vs. the max of other eight tissues under NT conditions. The x-axis indicates the max of transcript abundances in the other eight tissues (except pollen and anthers), and the y-axis indicates the average of transcript abundance in the 13 anthers, with each point representing a unique gene model. Blue points and orange points represent the anther-specific genes (ASGs) and anther-nonspecific genes, respectively. Pollen-abundant genes (AAGs) are located above the redline (y = log_2_ 300).

**Figure 3 ijms-23-03378-f003:**
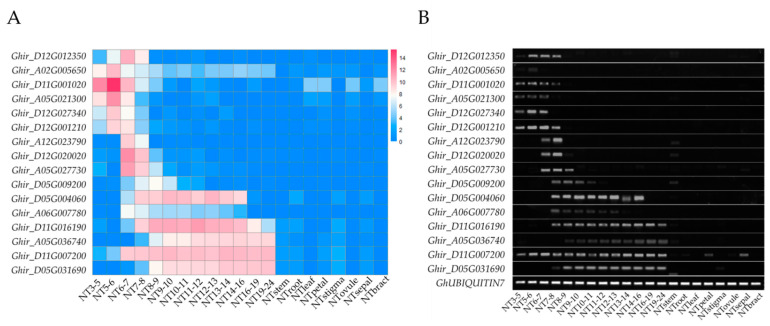
The expression pattern of 16 anther-specific genes (ASGs) in H05 tissues. (**A**) The expression pattern of 16 ASGs were verified by RNA-seq; (**B**) The expression pattern of 16 ASGs was verified by RT-PCR. The *GhUBIQUITIN7* gene was used as a reference gene; In (**A**,**B**), the row names represent the gene ID of 16 ASGs, and the column names represent 21 tissues of H05.

**Figure 4 ijms-23-03378-f004:**
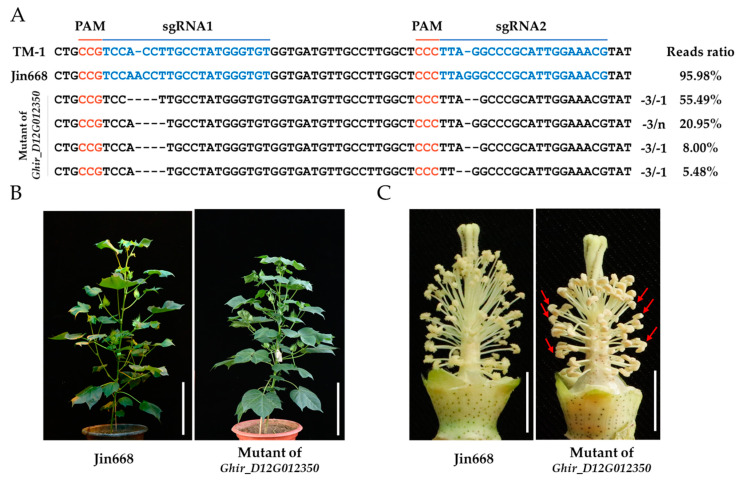
Genome editing of *Ghir_D12G012350* causes indehiscent anther in cotton. (**A**) Genome editing in the mutant of *Ghir_D12G012350* is exhibited. The sgRNA target sites and the PAM (protospacer adjacent motif) regions are shown in blue and orange, respectively. TM-1—the reference genome. Jin668—the wild type (WT); (**B**) The WT and mutant plants are shown. Scale bars = 20 cm. (**C**) The flowers of WT and mutant plants are shown, with petals removed. The red arrows represent the indehiscent anthers. Scale bars = 1 cm.

**Figure 5 ijms-23-03378-f005:**
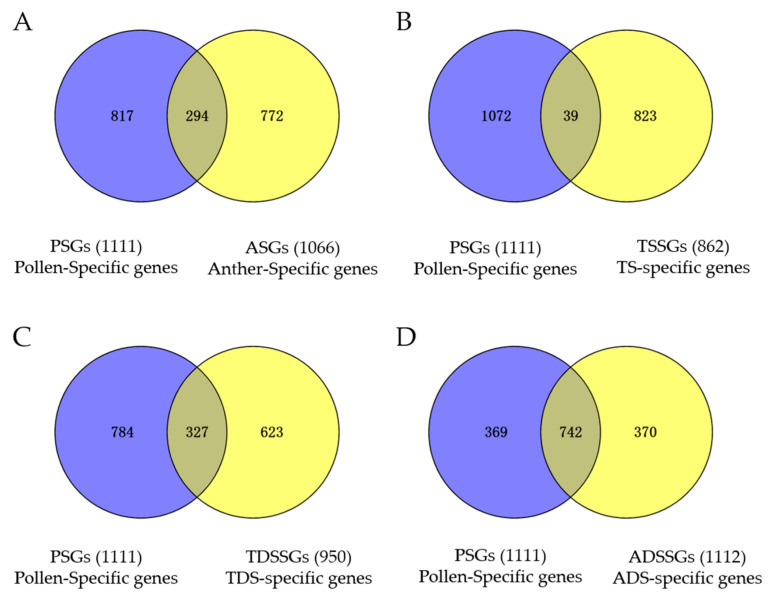
Venny plot analysis of APSGs, TSSGs, TDSSGs, and ADSSGs. (**A**) Identification of anther- and pollen-specific genes (APSGs) by using 1111 pollen-specific genes (PSGs) and 1066 anther-specific genes (ASGs); (**B**) Identification of tetrad stage- and pollen-specific genes (TSSGs) by using 1111 pollen-specific genes (PSGs) and 862 tetrad stage-specific genes (TSSGs); (**C**) Identification of tapetal degradation stage- and pollen-specific genes (TDSSGs) by using 1111 pollen-specific genes (PSGs) and 950 tapetal degradation stage-specific genes (TDSSGs); (**D**) Identification of anther dehiscence stage- and pollen-specific genes (ADSSGs) by using 1111 pollen-specific genes (PSGs) and 1112 anther dehiscence stage-specific genes (ADSSGs); In (**A**–**D**) the overlapped sets represent APSGs, TSSGs, TDSSGs and ADSSGs, respectively.

**Figure 6 ijms-23-03378-f006:**
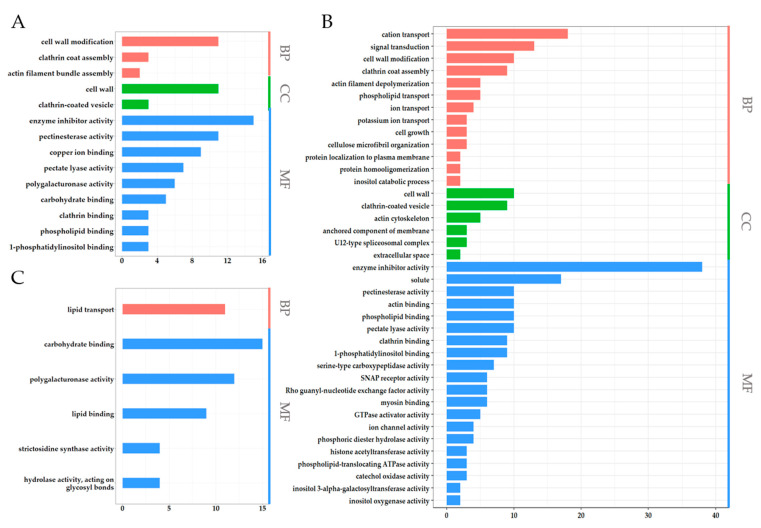
Gene ontology (GO) enrichment analysis of APSGs, PSGs, and ASGs in H05. (**A**) GO term enrichment analysis of APSGs; (**B**) GO term enrichment analysis of PSGs; (C) GO term enrichment analysis of ASGs; In (**A**–**C**), the x-axis represents the gene number assigned to a GO term, and the y-axis is the list of significantly enriched GO terms. These significantly enriched GO terms were selected based on a *p*-value < 0.05 and a *q*-value < 0.2. GO terms of the categories of biological processes, cellular components, and molecular functions are shown in red, green, and blue, respectively.

**Figure 7 ijms-23-03378-f007:**
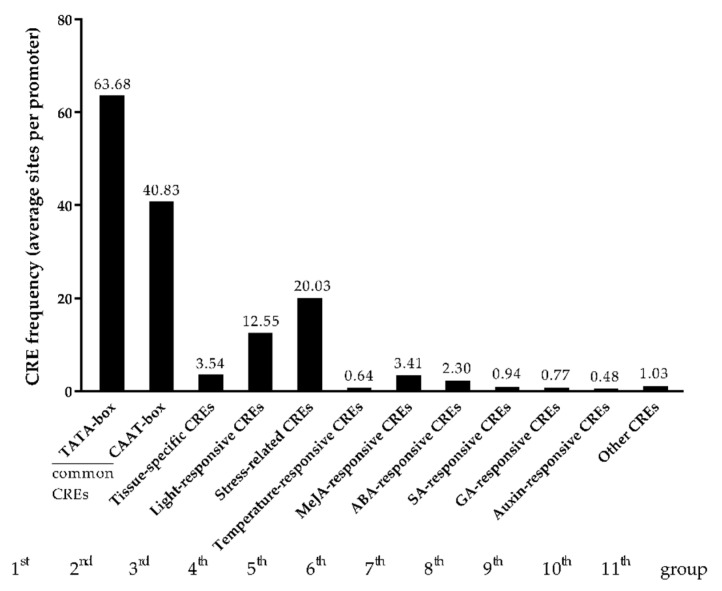
Classification of CREs identified in the promoters of APSGS genes; the x-axis represents the 11 groups of CREs classed by function; the y-axis indicates the CRE frequency: average sites per promoter.

**Figure 8 ijms-23-03378-f008:**
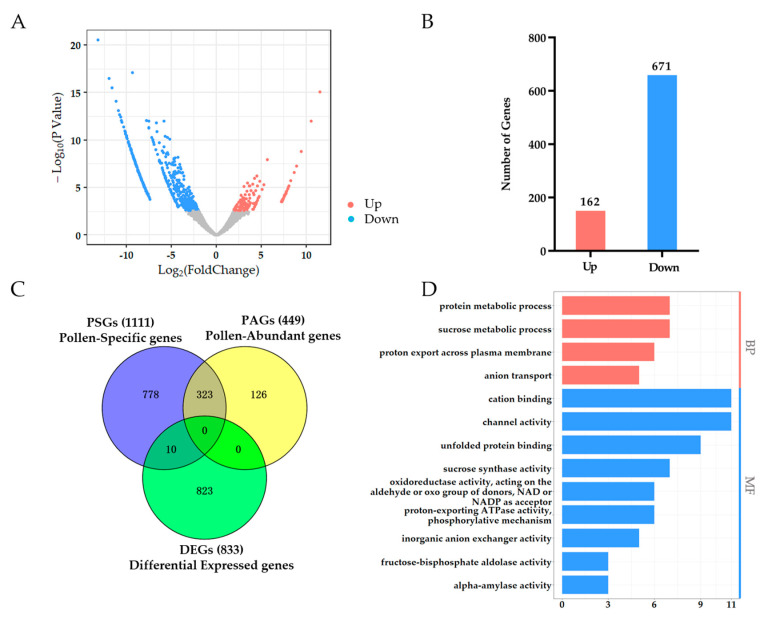
Volcano plot, Venny plot, and GO enrichment analysis of differentially expressed genes (DEGs). (**A**) Volcano plot of DEGs, the x-axis indicates the value of log_2_ (FoldChange) and the y-axis indicates the value of −log10 (*p*-value); (**B**) Number of up-regulated genes and down-regulated genes; (**C**) Venny plot of PSGs, PAGs, and DEGs; (**D**) GO enrichment analysis of DEGs, the x-axis represents the gene number assigned to a GO term and the y-axis is the list of significantly enriched GO terms.

**Figure 9 ijms-23-03378-f009:**
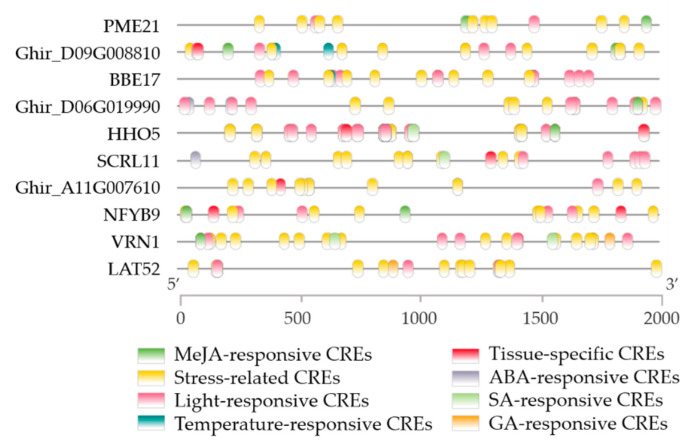
The distribution of CREs in the 2-kb promoters of 10 shared genes in PSGs and DEGs. The distribution of CREs was drawn by software TBtools, and 8 CREs were shown (not including the TATA box and CAAT box).

**Table 1 ijms-23-03378-t001:** The transcriptional levels of expressed genes (EGs), top 300 EGs, and top 1000 EGs in H05 tissue types.

Tissues ^a^	Number of EGs ^b^	Average Transcription of EGs (TPM) ^c^	Average Transcription of Top 300 EGs (TPM) ^d^	Average Transcription of Top 1000 EGs (TPM) ^e^	Percentage of Top 300 EGs in Transcriptome ^f^	Percentage of Top 1000 EGs in Transcriptome ^g^
NTpollen	12,005	82.8	2489.1	893.2	75.1%	89.9%
NT3-5	38,380	25.9	779.4	396.8	23.5%	39.9%
NT5-6	40,030	24.9	1000.2	421.9	30.1%	42.3%
NT6-7	42,867	23.2	702.5	336.3	21.2%	33.8%
NT7-8	41,853	23.8	789.6	373.7	23.8%	37.5%
NT8-9	41,059	24.2	907.3	409.5	27.4%	41.2%
NT9-10	39,798	25	1139.2	479	34.3%	48.1%
NT10-11	38,947	25.5	1081.7	470	32.7%	47.3%
NT11-12	37,567	26.5	1253.3	522.2	37.8%	52.5%
NT12-13	37,389	26.6	1236.4	516.8	37.3%	52.0%
NT13-14	36,763	27.1	1319.6	541.6	39.7%	54.4%
NT14-16	35,056	28.4	1343.8	556.3	40.5%	55.9%
NT16-19	30,917	32.2	1506.1	613.3	45.4%	61.6%
NT19-24	32,088	31	1370.7	574.4	41.3%	57.7%
NTroot	35,003	28.4	856.7	431.1	25.9%	43.4%
NTstem	41,397	24.1	814.8	383.6	24.5%	38.4%
NTleaf	38,662	25.7	1416.4	551.6	42.8%	55.5%
NTbract	34,974	28.4	1138.7	508.3	34.4%	51.2%
NTsepal	37,484	26.6	866.6	418.2	26.1%	41.9%
NTpetal	35,495	28	822.8	419.5	24.8%	42.2%
NTstigma	38,303	26	760.6	378.7	22.9%	38.0%
NTovlue	40,855	24.4	634.1	322.9	19.1%	32.4%

^a^ Twenty-two squenced tissues of cotton H05 under normal temperature (NT) conditions: mature pollen at anthesis under NT (NTpollen), anther in 3–5 mm bud under NT (NT3-5), anther in 5–6 mm bud under NT (NT5-6), anther in 6–7 mm bud under NT (NT6-7), anther in 7–8 mm bud under NT (NT7-8), anther in 8–9 mm bud under NT (NT8-9), anther in 9–10 mm bud under NT (NT9-10), anther in 10–11 mm bud under NT (NT10-11), anther in 11–12 mm bud under NT (NT11-12), anther in 12–13 mm bud under NT (NT12-13), anther in 13–14 mm bud under NT (NT13-14), anther in 14–16 mm bud under NT (NT14-16), anther in 16–19 mm bud (NT16-19), anther in 19–24 mm bud (NT19-24), root under NT (NTroot), stem under NT (NTstem), leaf under NT (NTleaf), bract under NT (NTbract), sepal under NT (NTsepal), petal under NT (NTpetal), stigma under NT (NTstigma), and ovlue under NT (NTovlue). ^b^ The number of expressed genes (EGs) whose transcriptional level is higher than 1 TPM (Transcripts Per Kilobase of exon model per Million mapped reads). ^c^ The average transcription of EGs is the average TPM value of all EGs. ^d^ or ^e^ The average transcription of the top 300 EGs or 1000 EGs is the average TPM value of top 300 EGs or 1000 EGs. ^f^ or ^g^ The percentage of top 300 EGs or 1000 EGs in transcriptome is the percentage that the transcripts of top 300 or 1000 EGs occupied the whole transcriptome.

## Data Availability

The RNA-sequencing raw data were deposited with the NCBI under Sequence Read Archive (SRA) under accession number PRJNA812406.

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
