# Peer review of "Rapid Identification of Pollen- and Anther-Specific Genes in Response to High-Temperature Stress Based on Transcriptome Profiling Analysis in Cotton"

_ijms, 2022, doi:10.3390/ijms23063378_

Round 1

Reviewer 1 Report

Manuscript ijms-1617340 “Rapid identification of pollen- and anther-specific genes in response to high-temperature stress based on transcriptome profiling analysis in cotton” by Zhang et al. showed an interesting study about gene expression associated to pollination in cotton.

However instead of the good analysis performed, the present version of the manuscript presents some deficiencies which must be revised before publication. Manuscript is confused in the description of the material assayed and the methodology. Description of results is also very poor.

This manuscript IS NO ACEPTABLE to be published in International Journal of Molecular Science.

Major points for the REJECTION of the manuscript:

Objectives of the work must be clarified in a separated paragraph without references. Key points, the phenotyping protocol assayed (if it is an objective) and the need to clarify pollination in cotton in a context of high temperature stress.

Results description is very limited. Pollen- and anther interaction must be clarified.

Raw Data from RNA-Seq experiments should be added as a SRA file deposited in the NCBI database.

qRT-PCR analysis should be also clarified indicating the nature of the assayed technical and biological samples. In my opinion RNA-Samples must come from a different assay, not exactly the same that the RNA-Seq assay. This question must be clarified. In addition, the selection of the six candidate genes must be justified. If it is possible a higher number of genes to validate RNA-Seq data should increase the robustness of the experiment. Authors must incorporate the RNA-Seq reads results and the correlation coefficient between qPCR and RNA-Seq data. qPCR analysis is the only way to validate RNA-Seq data and to establish correct biological hypothesis.

Plant Material must be clarified and better explained. Cotton genotype assayed must be better described.

Phenotype evaluation of samples must be included. In theory the phenotype is the pollen- and anther interaction.

Samples assayed in the RNA isolation must be clarified also clarifying the pollen- and anther interaction.

Number of biological and technical replications must be clarified in the RNA-Seq and qPCR experiments. In addition, authors must be clearly indicated in the biological samples in bot experiments are the same or are from different experiments.

Conclusions of the work are weak and should be also an artefact taking in account the not validated phenotyping protocol. Main implications of the obtained data from a production and breeding point of view should be added.

Reviewer 2 Report

It is quite interesting and important research. Authors significantly improve writing and paper organisation if compared with original submission (ijms-1464095).

Lines 66-67: do not forget to mention that pollen development form meiosis till mature pollen is a series of very rapid chnages in chromatin organization through different histone modification. This must be at least mentioned in the introduction and discussion. In ideal case if authris in future can study effcet of HAT on this process by precise analysis of chromatin orgsanization in response to HAT as upstream regulator of pollen fertility.   

Lines 242-246: is an important confirmation of the results.   

Lines 421-428: it is well written, but need to mention that pollen developmnet is a rapid chnages in histone modification as a stress-response signal. Please, discuss how chnages in gene expression you have found linked with epignetic of the pollen development.    

Round 2

Reviewer 1 Report

Thanks for all the amendments performed in this re-submitted version of the manuscript.
I think it is a much better work than before.
However, some revisions are necesary before publication:

Clarify Objective of the work in a separated parragraph

Complete RNA-Seq description

Conclusions should be completed to be more concrete and less vague

Author Response

thanks

Reviewer 2 Report

Thank you,

it is OK now.

Plesae, add size of scale bar to figure 4.

Author Response

thanks

Round 3

Reviewer 1 Report

Authors have revised correctly the manuscript